# Regulatory Mechanism of Proanthocyanidins in Grape Peels Using vvi-miR828a and Its Target Gene *VvMYBPA1*

**DOI:** 10.3390/plants13121688

**Published:** 2024-06-18

**Authors:** Lingqi Yue, Jingjing He, Tian Gan, Songtao Jiu, Muhammad Khalil-Ur-Rehman, Kunyu Liu, Miao Bai, Guoshun Yang, Yanshuai Xu

**Affiliations:** 1College of Horticulture, Hunan Agricultural University, Changsha 410128, China; 18737690195@163.com (L.Y.); baimiao@hunau.edu.cn (M.B.); guoshunyang@aliyun.com (G.Y.); 2Department of Plant Science, School of Agriculture and Biology, Shanghai Jiao Tong University, Shanghai 200240, China; jiusongtao@sjtu.edu.cn; 3Horticultural Sciences, The Islamia University of Bahawalpur, Bahawalpur 62300, Pakistan; muhammad.khalil@iub.edu.pk

**Keywords:** grape, miRNA-seq, vvi-miR828a, *VvMYBPA1*, PA

## Abstract

Anthocyanins and proanthocyanidins are considered to be essential secondary metabolites in grapes and are used to regulate metabolic processes, while miRNAs are involved in their synthesis of anthocyanins and proanthocyanidins to regulate metabolic processes. The present research work was carried out to investigate the underlying regulatory mechanism of target genes in the grape cultivars ‘Italia’ and ‘Benitaka’. miRNA and transnscriptomic sequencing technology were employed to characterize both the profiles of miRNAs and the transcripts of grape peels at 10 and 11 weeks post flowering (10 wpf and 11 wpf). The results revealed that the expression level of vvi-miR828a in ‘Italia’ at 10 and 11 wpf was significantly higher than that in ‘Benitaka’. miRNA-seq analysis predicted *MYBPA1* to be the target gene of vvi-miR828a. In transcriptome analysis, the expression level of the *VvMYBPA1* gene in ‘Benitaka’ was significantly higher than that in ‘Italia’; in addition, the TPM values (expression levels) of *VvMYBPA1* and miR828a also showed an evident negative correlation. The determination of the proanthocyanidin (PA) content in ‘Italia’ and ‘Benitaka’ peels at 11 wpf demonstrated that the PA content of ‘Benitaka’ was significantly higher than that of ‘Italia’. The outcomes of RT-qRCR analysis exhibited that the expression levels of the *VdPAL*, *VdCHS*, *VdCHI*, *VdDFR*, *VdMYB5b*, *VdANR*, and *VdMYBPA1* genes related anthocyanin and proanthocyanidin pathways were reduced, while the expression levels of all of the above genes were increased after the transient expression of the *VvMYBPA1* vector into grape leaves. The results of the transient overexpression experiment of vvi-miR828a before the veraison period of strawberry fruits showed that vvi-miR828a can significantly slow down the coloration of strawberries. The vvi-miR828a negatively regulates the accumulation of proanthocyanidins in grape fruits by inhibiting the expression of *VvMYBPA1*.

## 1. Introduction

Anthocyanins not only contribute to fruit coloring but also offer health benefits by providing protection against diseases such as cancers, diabetes, and neurological disorders.

Anthocyanins and proanthocyanidins (PAs) belong to the class of flavonoids, which are divided into several subgroups including flavones, flavanols, flavanones, catechins, anthocyanins, and chalcones. Anthocyanins not only offer health nutritional benefits but are also associated with color development in fruits as colors, while proanthocyanidins provide astringency and bitterness in the fruits and prevent the picking of early or unripe fruits to avoid commercial yield loss [1,2,3,4]. Anthocyanins and proanthocyanidins are considered as essential metabolites and important components of processed agricultural products due to their nutritional value. These metabolites also provide protection from heart disease, cancer, and many other diseases due to their high medicinal importance [5,6].

The anthocyanins and PAs are regulated by similar metabolic pathways in grapes and synthesized from the amino acid phenylalanine through the phenylpropanoid metabolism pathway [6,7]. The flavonoids and stilbenes are generally associated with the phenylalanine pathway [8]. The biosynthesis of flavonoid compounds is initiated by one p-coumarol coenzyme A and three propylene glycol coenzyme A molecules, while p-coumarol coenzyme A is synthesized from phenolic acids [9]. Many other enzymes are involved in the phenylalanine pathway, including chalcone isomerase (CHI), flavonoid 3-hydroxylase (F3H), flavanol synthase (FLS), and dihydro flavanol reductase (DFR). Anthocyanin synthase (ANS)/glucose transferase (UFGT) and *MYBPA1* regulate the synthesis of proanthocyanidins by activating the expression of these genes or enzymes, which further promote the formation of naringenin, dihydro flavanols, flavanols, and anthocyanins [10,11,12]. Several metabolic changes occur during the plant growth and development period. The fruit development cycle in grapes is a double S-curve; the first cycle starts from the growth and development period, from flowering and fruit setting to veraison, while second cycle starts from veraison to ripening [13]. The biosynthesis of proanthocyanidins usually starts after flowering and fruit setting at 3~9 weeks post grape fruit anthesis, and the expression of *MYBPA1, ANR*, and *LAR* genes promotes the synthesis of proanthocyanidins that gradually increase in the peel and seeds. Proanthocyanidin accumulation initiates after fruit set and attains its peak during the veraison period. The anthocyanins begin to accumulate significantly after the veraison period, and the expression of *MYBA1* gradually increases until fruit ripening [14,15].

Previous studies have revealed the importance of enzymes and structural genes in the flavonoid biosynthetic pathway and the regulatory role of some key genes/transcription factors in grapes. Among these transcription factors (TFs), unique ones are MYB, bHLH, and WD40. They form a transcription complex protein and are involved in regulating the synthesis of anthocyanins and proanthocyanidins. The MYB transcription factor is considered important among these three TFs [16,17]. The MYB gene/transcription factors have been shown to be involved in the control of flavonoid synthesis; among these TFs, *VvMYB5a* and *VvMYB5b* are involved in the general flavonoid metabolism pathway [13,18], *VvMYBPA1* is involved in the PA synthesis pathway [19], and *VvMYBA1* and *VvMYBA2* are involved in the anthocyanin synthesis or biosynthesis pathway [20,21,22,23].

Recent literature has elucidated the role of microRNA (miRNAs) in the regulation the flavonoid biosynthetic pathway [24]. Plant miRNA is a type of short non-coding RNA (21–24 nucleotides) that regulates the post-transcriptional expression of genes [25]. It plays a regulatory role in plant growth and development, as well as instress response, and secondary metabolism [26]. miRNA is typically initiated by its own promoter and transcribed by RNA polymerase II. Initially, it produces the primary transcript pri-miRNA; which is then, recognized and cleaved by DCL1 (RNase III endoribonuclease DICER-like 1). his process forms precursor-miRNA (pre-miRNA), characterized by a stem-loop structure and specific folding free energy. The pre-miRNA is subsequently cleaved again by DCL1 to form a miRNA–miRNA* double-stranded complex. Subsequently, HEN1 (S-adenosylmethionine-dependent methyltransferase HEN1) methyltransferase methylates the 3′ terminal base of the double strand to protect it from degradation. Methylated miRNA double strands are exported from the nucleus to the cytoplasm by the plant export protein HASTY. In the cytoplasm, one of the mature miRNA strands is degraded, and the other strand enters the RNA-induced gene silencing complex RISC and combines with the Argonaute protein to form the RISC silencing complex [27,28,29]. miRNA mainly cleaves or inhibits the target gene by complementary pairing with the sequence of the target mRNA, which means that when the sequence of the miRNA and the target mRNA are completely complementary, the miRNA cleaves the target mRNA, leading to its degradation. When the sequences of miRNA and target mRNA are not perfectly complementary, the translation of target gene mRNA is inhibited [30,31]. In Arabidopsis, apple, and tomato, the *SPL*-miR156 module (*SPL* gene) regulates anthocyanin synthesis and is highly conserved in plants. *SPL* affects anthocyanin biosynthesis by destroying the MBW complex [32,33]. *SPL*-miR156 has also been found to regulate the synthesis of anthocyanins in grapes, pomegranates, blueberries, and other fruit trees [34,35,36]. Many other miRNAs, such as miR157a, miR159, miR162, miR167a, miR172, miR395, miR396, miR396a, miR828, miR858a, etc., were also found to regulate the structural genes *4CL, CHI, F3H, LDOX,* and *UFGT,* as well as WD40, bHLH, and MYB transcription factors in the flavonoid biosynthetic pathway, thereby affecting the synthesis of secondary metabolites such as anthocyanins, proanthocyanidins, and flavanols [15,24]. The proanthocyanidins in persimmons are formed through the activation of laccase enzyme polymers of flavan-3-ols stored in the vacuoles. The laccase gene *DkLAC2* is involved in proanthocyanidin biosynthesis and is regulated by miR397. After overexpression of DkmiR397, the laccase gene *DkLAC2* is inhibited in persimmon, and the synthesis of proanthocyanidins is reduced [37].

In grapes, vvi-miR828 and vvi-miR858 inhibit anthocyanin synthesis by negatively regulating *VvMYB114* [38,39]. However, there is barely any research on the regulation of proanthocyanins by miRNAs in grapes. In this study, the fruit peels of ‘Italia’ and ‘Benitaka’ bud sport cultivar before the veraison period (10 weeks post flowering) and at the veraison period (11 weeks post flowering) were used as test materials by conducting small RNA and transcriptome sequencing to explore candidate miRNAs that regulate proanthocyanidins and predict their target genes. The effect of vvi-miR828a on anthocyanin synthesis was studied by detecting the PA content in the peels of ‘Italia’ and ‘Benitaka’ and transiently expressing candidate miRNA (vvi-miR828a) in strawberry fruits during the veraison period. By transiently expressing vvi-miR828a and *VvMYBPA1* in spine grape (*Vitis davidii*) leaves and comparing the expression of key genes in the anthocyanin and proanthocyanidin biosynthetic pathways with RT-qRCR, we observed the negative regulation of vvi-miR828a on its target genes and its inhibitory effect on proanthocyanidin synthesis. The outcomes of our study will ultimately be helpful in understanding the regulatory role of vvi-miR828a in PA biosynthesis as well as in proposing future research to comprehend the mechanism involved in PA biosynthesis.

## 2. Results

### 2.1. Sequencing Depth and Quality Control of Small RNA

The number of raw reads obtained from each sample in the 10 wpf and 11 wpf periods of ‘Italia’ and ‘Benitaka’ grape cultivars was 144,895,190 (It10), 113,765,862 (It11), 154,764,914 (Be10), and 181,890,944 (Be11). After filtering the data and removing low-quality fragments (reads), including those with incorrect adapters and those with plot Ns, the clean reads were 144,890,936 (It10), 113,750,206 (It11), 154,760,108 (Be10), and 181,878,014 (Be11). Fragments with a length of <18 nt were removed, and each sample obtained fragments ranging from 58.4% to 70.6% for comparison with the reference genome.

The total length of small RNAs measured via comparison was 18~25 nt. Among the four samples, the most abundantly expressed small RNA was 21 nt in length followed by small RNA of 24 nt in length (Figure 1). Using the miRBase and Rfam databases, we compared the fragments that aligned with the reference genome in order to analyze small RNAs and to screen out known non-coding RNAs (snoRNA, snRNA, tRNA, and RNAs). The composition of all non-coding RNAs was similar among the four samples, with only minor differences occurring within each sample (Table 1). Among the components of all non-coding RNAs, known miRNAs account for the largest average proportion of read fragments, reaching an average of 21.62% (Table 1), followed by rRNA (21.44%); the proportions of other RNAs are the newly predicted 5.01% for miRNA, 0.43% for tRNA, 0.15% for snoRNA, and 0.015% for snRNA.

### 2.2. Identification of Known and New miRNAs

Through sequence alignment analysis, a total of 306 miRNAs were identified in the It10 sample (including 127 known miRNAs and 179 new miRNAs); a total of 310 miRNAs were identified in the It11sample (including 119 known miRNAs and 191 new miRNAs); a total of 330 miRNAs were identified in the Be10 sample (126 known miRNAs and 204 new miRNAs); and 332 miRNAs were identified in the Be11 sample (including 135 known miRNAs and 197 new miRNAs). In total, 267 miRNAs were detected between the two samples of ‘Italia’, while 287 miRNAs were detected in the two samples of ‘Benitaka’; in the comparative analysis of ‘Italia’ and ‘Benitaka’, 248 miRNAs (Figure 2A) were detected. There were 265 detected miRNAs in the four samples, and the following eight were specifically expressed in ‘Italia’: vvi-miR171c, vvi-miR171d, vvi-miR171j, vvi-miR156e, vvi-miR399e, 13_23166, 14_28417, and Un_40751. A total of 48 genes were specifically expressed in the ‘Benitaka’ cultivar (Figure 2B). The expression levels of all specifically expressed miRNAs were relatively low.

The identified known miRNAs that were identified are divided into 31 conserved miRNA families, most of which are found in a variety of plants; in addition to the conserved miRNA families, sequencing also detected some known but less conserved miRNA families including the following: MIR3623, MIR3624, MIR3625, MIR3626, MIR3627, MIR3628, MIR3629, MIR3632, MIR3633, MIR3634, MIR3635, MIR3636, MIR3637, MIR3638, MIR3639, and MIR3640. Among the known conserved miRNA families, MIR169_2 had the most members, with 12, followed by the MIR156, MIR166, and MIR159 families, with 7–8 members; other families have one to six members. Members of the same family of non-coding miRNAs show different expression profiles; for example, the expression of most miRNAs of the MIR169 family in ‘Benitaka’ decreases with fruit development, while the expression of miR169t increases with fruit development.

### 2.3. Expression Trends of All Obtained miRNAs

The relative expression of miRNA was measured in TPM. A notable variation was observed in the expression pattern of each miRNA in the different samples. Some miRNAs had a high expression in all samples, while some had the opposite expression. A total of 27 miRNAs had a TPM value greater than 500. In order to gain a deeper understanding about the expression of miRNA in the different samples, the k-means clustering method was used to conduct a sub-clustering analysis of all miRNAs through the transformation value of Log10 (TPM + 1); all miRNAs were divided according to their expression trend. The miRNAs were divided into eight gene clusters (Appendix A). Among the eight gene clusters, three showed consistent expression trends in the four samples of ‘Italia’ and ‘Benitaka’, while the other five gene clusters showed different expression trends in the two cultivars. Overall, diverse expression trends of miRNAs among the four samples were noticed.

The results of the miRNA gene cluster analysis showed that the number of members of the above eight gene clusters were 20, 8, 20, 22, 8, 20, 15, and 11, respectively. The eighth gene cluster contains 11 miRNAs; during the fruit development period of ‘Italia’ grapes, the expression levels of these 11 miRNAs increased significantly, while in the ‘Benitaka’ cultivar, the expression levels did not change significantly. The second and fifth gene clusters each contain eight miRNAs; their expression in the ‘Benitaka’ cultivar shows a gradual decreasing trend with fruit development, while the ‘Italia’ cultivar shows the opposite trend. The fourth and sixth gene clusters contain 22 and 20 miRNAs, respectively, and their expression levels increased significantly with fruit development in the ’Benitaka’ cultivar, while the opposite trend was shown in the ’Italia’ cultivar. The first and the third gene clusters comprise 20 miRNAs, and the expression trends in the two cultivars were consistent. The seventh gene cluster contains 15 miRNAs in the ’Benitaka’ cultivar, and these miRNAs increased significantly with fruit development. Their expression was not obvious before the veraison period but increased significantly after it. In contrast, in the ‘Italia’ variety, the expression trends of these miRNAs showed an insignificant decrease with fruit development. In particular, miRNAs such as vv-miR171c, vv-miR171d, vv-miR171j, and vv-miR399e were not expressed in the ‘Italia’ cultivar but were significantly reduced with fruit development in the ‘Benitaka’ cultivar.

The predicted target genes were detected in small numbers as specifically expressed miRNAs in both the ‘Italia’ and ‘Benitaka’ cultivars; the target genes of these miRNAs were annotated as *SPL* (scale promoter binding protein), laccase, auxin response factor, E3 ubiquitin ligase, WRKY transcription factor, ultraviolet receptor UVR8, cytochrome P450, UDP-glycosyltransferase, copper transporter, ethylene response factor, and bZIP transcription factor, respectively.

### 2.4. Differential Expression of miRNA in Two Cultivars

In total, 161 differentially expressed miRNAs were present in the Be10 and Be11 samples, including 72 up-regulated and 89 down-regulated miRNAs. On the other hand, 90 differentially expressed miRNAs were observed between the It10 and It11 samples (20 were up-regulated and 70 were down-regulated). Additionally, there were 181 differentially expressed miRNAs between the It10 and Be10 samples, including 119 up-regulated and 62 down-regulated miRNAs. There were also 105 differentially expressed miRNAs between the It11 and Be11 samples (63 were up-regulated and 42 were down-regulated) (Figure 3A). Among the differentially expressed miRNAs in the four samples, the number of up-regulated miRNAs in ‘Benitaka’ was greater than that in ‘Italia’, in which most of the differentially expressed miRNAs showed a down-regulated expression. In addition, some miRNAs were also found to be specifically expressed in ‘Italia’ and ‘Benitaka’ at the 11 wpf stage (Figure 3B).

### 2.5. Target Gene Prediction of miRNAs

A total of 4356 target genes were predicted; out of these genes, 73 were differentially expressed miRNAs(|log_2_FC| > 1). vvi-miR160a had the lowest number of target genes (49), while vvi-miR397a had the highest number (182). The results depicted that the same miRNA is involved in multiple biological functions during plant growth and development.

All predicted target genes were subjected to GO enrichment and KEGG analysis; a total of 4356 target genes were annotated into three functional categories: biological processes, cellular components, and molecular functions. The three functional categories have 10, 3, and 7 GO items, respectively. The majority of genes were observed in metabolic processes, protein binding, and cellular component processes. In the KEGG pathway analysis, target genes were enriched in 19 pathways, among which, genes related to metal ion fixation, metal ion transport, and phenylalanine metabolism were highly enriched (Appendix A).

The results of cluster analysis showed that among the differentially expressed miRNAs of the samples It11 and Be11, the expression of vvi-miR828a, vvi-miR156e, and vvi-miR399a gradually increased with fruit development in ’Italia’; in ‘Benitaka’, its expression gradually decreased during fruit development. The expression of vvi-miR408, vvi-miR2111-5p, vvi-miR479, vvi-miR160c, and vvi-miR477b-3p showed the opposite trend (Appendix A). For target gene prediction, it was found that the target genes of these miRNAs were annotated as ethylene response factors, laccases, ABC transcription factors, sucrose transporters, glutathione transferases, bHLH transcription factors, WRKY transcription factors, and MYB transcription factors.

The target gene enrichment analysis indicated that vvi-miR828a targeted the most MYB transcription factors; therefore, it was selected for further analysis. Among all the target genes of vvi-miR828a, it was observed that the level of the *VvMYBPA1* gene was relatively and significantly different in expression levels between the two cultivars (Table 2).

Our previous laboratory report studied the transcript expression levels in the peels of ‘Italia’ and ‘Benitaka’ grapes at 10 and 11 wpf [40]; it was found that there was an obvious negative correlation between them (Figure 4A). In four sample tissues, the expression level of vvi-miR828a was detected, and the results were consistent with the RNA-seq results. The expression level of vvi-miR828a was not detected in the Be10 sample; in the Be11, It10, and It11 samples, its expression was detected. The qRT-PCR results were consistent with this. Among all tissue samples, vvi-miR828a expression was highest in the peel of the ‘Italia’ cultivar at 11 weeks post flowering (Figure 4B). After measuring the PA content in the peel of It11 and Be11 at 11wpf, the results indicated a higher PA content in ‘Benitaka’ (0.151 mg/mL) compared to ‘Italia’ (0.017 mg/mL). The PA contents significantly differed in the peels of the two cultivars at 11 wpf (Figure 4C).

### 2.6. Verification of Dual-Luciferase Assay between vvi-miR828a and Target Gene VvMYBPA1

vvi-MIR828a was constructed into the pSAK277 vector as an effector; the target gene *VvMYBPA1* and its mutation target sequence were constructed into the 3’ end of the pGreenII 0800-LUC vector as a reporter (Figure 5A,B). OX-miR828a (OX, overexpressed) and the target gene *VvMYBPA1* were mixed and injected into tobacco leaves. The results of the dual-luciferase experiments showed that after injection, the expression of the LUC reporter gene was significantly inhibited. miR828a targeted *VvMYBPA1*, which was consistent with the results of target gene prediction (Figure 5C), indicating that miR828a targets and negatively regulates *VvMYBPA1*.

### 2.7. Effects of Transient Expression of miR828a and VvMYBPA1 on the Expression of Genes Related to Anthocyanin and Proanthocyanidin Synthesis

After the transient overexpression of vvi-miR828a, the expression levels of the *VdPAL, VdCHS, VdCHI, VdDFR, VdMYB5b, VdANR*, and *VdMYBPA1* genes were all significantly reduced. On the other hand, after the transient overexpression of *VvMYBPA1*, the expression of these genes significantly increased; this result was significantly different from the results of the transient overexpression of vvi-miR828a (Figure 6A–G). It is speculated that vvi-miR828a inhibits the expression of genes related to anthocyanin and proanthocyanidin biosynthesis. A bacterial solution containing the vectors of miR828a and *VvMYBPA1* was injected into the leaves of *Vitis davidii*, and the contents of proanthocyanidins were detected in the leaves 5 days later. The results showed that miR828a significantly inhibited the synthesis of proanthocyanidins, while *VvMYBPA1* significantly increased the synthesis of proanthocyanidins (Figure 6H).

### 2.8. Transient Expression to Verify the Biological Function of vvi-miR828a

After continuous observation of strawberry fruits that transiently overexpressed vvi-miR828a but did not change color, the results indicated that the coloration of strawberry fruits was inhibited after the transient overexpression of vvi-miR828a, which slowed down strawberry fruit coloration compared to the control (Figure 7A). This result indicates that vvi-miR828a may affect the accumulation of anthocyanins in fruits. A bacterial solution containing vectors of miR828a and *VvMYBPA1* was injected into the strawberry fruits, respectively. The content of proanthocyanidins in strawberry fruits after the overexpression of miR828a and *VvMYBPA1* was detected 5 days post inoculation. The results showed that miR828a significantly inhibited the synthesis of proanthocyanidins, *VvMYBPA1* significantly increased the synthesis of proanthocyanidins (Figure 7B), and miR828a inhibited the biosynthesis of proanthocyanidins.

## 3. Discussion

In grapes, anthocyanins and PAs are regulated by identical metabolic pathways; many structural genes and transcription factors in these pathways have been analyzed. *VvMYBA1* and *VvMYBA2* are considered as key genes, which specifically regulate anthocyanin biosynthesis in grape berries; *VvMYBPA1* and *VvMYBPA2* mainly regulate the biosynthesis of proanthocyanidins in grapes [41]. *VvMYB5a* and *VvMYB5b* are also involved in the synthesis of grape anthocyanins and proanthocyanidins [42]. miRNA is a type of endogenous non-coding small RNA that negatively regulates gene expression post-transcriptionally and participates in plant signal transduction, hormone response, secondary metabolism, and other processes.

Some studies on miR828 and miR858 functioning and characterization have been reported in plants; their main biological function is to participate in the regulation of the synthesis and accumulation of plant anthocyanins and proanthocyanidins. In tomatoes, miR828 negatively regulates the accumulation of anthocyanins; in fruits the expression of anthocyanins and related enzyme genes in the synthesis pathway decreased in the T1 generation of miR828 transgenic tomatoes [43]. In grapes under root-limited cultivation, it was observed that the anthocyanin content in the fruits was higher. Previous sequencing reports revealed that the expression of miR828 was decreased under root-limited cultivation conditions (treatment). The over expression of miR828 in Arabidopsis showed light-colored leaf types, consistent with the phenotype of grape berries in controls, suggesting that miR828 acts as a repressor of anthocyanin biosynthesis [38]. A recent report stated that miR828 inhibits the intensity of petal color during the growth and development of pink strawberries [44]. Our results indicated that miR828 may be involved in the regulation of PA biosynthesis (Figure 4 and Figure 6). miRNA has also been found to be involved in regulating the biosynthesis of PA in apples, persimmons, cotton [45], and leguminous plants [46]. Among them, miR858 negatively regulates PA accumulation in apple peel by targeting *MdMYB9/11/12* [47]. An miRNA (miR858b) that negatively regulates proanthocyanidin synthesis was identified in persimmons; it can target *MYB19* and *MYB20* by transiently expressed miR858b in persimmon fruits, and the results showed that it inhibited the PA content in persimmons. The results of the transient expression of *MYB19* and *MYB20* showed that these genes increased the PA content in persimmons [48].

The results of miRNA-seq found that the expression level of vvi-miR828a gradually increased with berry development in ‘Italia’ grapes, while the expression level demonstrated the opposite trend in the ‘Benitaka’ grape berries; there were significant differences in the grape berries of the two cultivars before and after the veraison stage. The target gene prediction results demonstrated that most of the target genes of vvi-miR828a are MYB transcription factors, including *MYBPA1*, *MYB5b*, etc. Among them, *MYBPA1* has the highest expression level [40]. These transcription factors are involved in regulating the synthesis of flavonoids (including anthocyanins, proanthocyanidins, flavonols, flavones, etc.). Therefore, determining whether or not vvi-miR828a regulates *MYBPA1* became the main focus of this study.

Variations in the expression of *VvMYBPA1* in ‘Italia’ and ‘Benitaka’ cultivars at 10 and 11 wpf were noticed, and there was a significant negative correlation in the expression of vvi-miR828a at the 10 and 11wpf of the two cultivars. The expression of *VvMYBPA1* can regulate early fruit development and PA accumulation in seeds. The transient expression of *VvMYBPA1* in grape leaves revealed that the PA content in grape leaves increased significantly. *VvMYBPA1* can activate the expression of *VvLAR, VvANR*, and several flavonoid pathway genes, but cannot activate *VvUFGT*. *VvMYBPA1* may specifically regulate grape PA biosynthesis [19,41]. PA content in the peel of the ‘Italia’ and ‘Benitaka’ cultivars was detected at 11 wpf; the results indicated that the PA content in the peel of the ‘Benitaka’ variety was significantly higher than that of the ‘Italia’ variety. A vector bacterial solution containing miR828a and the target gene *VvMYBPA1* was mixed at a ratio of 9:1 and was injected into tobacco leaves. The results showed that, compared with tobacco leaves injected with ‘miR828a + the target gene *VvMYBPA1* CDS mutant sequence’, the reporter gene LUC/REN value of the ‘miR828a + normal *VvMYBPA1* CDS sequence (non-mutated sequence)’ was significantly reduced, which is consistent with the predicted results of miR828a targeting the negative regulation of *VvMYBPA1* (Figure 5C). It is speculated that miR828a may be involved in the regulation of grape proanthocyanidin biosynthesis.

The experimental results of the transient expression of vvi-miR828a during the veraison stage of strawberry fruits showed that vvi-miR828a significantly slowed down the coloration of strawberry fruits (Figure 7), indicating that vvi-miR828a may also negatively regulate anthocyanin biosynthesis. The results of the transient expression of vvi-miR828a and *VvMYBPA1* experiments, as well as qRT-RCR experiments in grape leaves, showed that the expression of the *VdPAL, VdCHS, VdCHI, VdDFR, VdMYB5b, VdANR,* and *VdMYBPA1* genes related to the biosynthesis of anthocyanins and proanthocyanidins was significantly reduced; contrarily, after the transient expression of *VvMYBPA1*, the expression of these genes increased significantly, which was significantly different from the transient overexpression of vvi-miR828a (Figure 6). vvi-miR828a inhibits the expression of anthocyanin and proanthocyanidin synthesis genes, which may affect the accumulation of anthocyanins and proanthocyanidins in grape berries. Therefore, vvi-miR828a may have a bidirectional effect in regulating grape anthocyanins and proanthocyanidins. In the early stages of grape fruit development, the expression of vvi-miR828a was very low, and its regulatory effect was not significant; after fruit development, its expression was increased.

One gene, *MYB5b*, which was involved in anthocyanin synthesis, was also exhibited in the prediction results of miR828a target genes in this study. In grape berries, *VvMYB5b* can activate genes for anthocyanin biosynthesis (except *VvUFGT*, which cannot be activated), promoting the accumulation of anthocyanins. At the same time, *VvMYB5b* can also activate the expression of the *VvLAR* and *VvANR* genes that play an important role in the regulatory mechanism of PA biosynthesis [13]. *MYB5b* activates several genes of the flavonoid biosynthetic pathway during the ripening of grape berries. Changes in *MYB5b* expression also affect flavonoid biosynthesis during this developmental stage. The *VvMYB5b* gene was transgenically overexpressed in petunias, and it was found that *VvMYB5b* induced the accumulation of anthocyanins. *MYBPA1* and *MYB5b* can activate genes in the flavonoid biosynthetic pathway [19], thereby affecting anthocyanin and PA content. In grapes, in addition to targeting and regulating *VvMYBPA1*, miR828a may also target *MYB5b*; therefore, the relationship between miR828a and *VvMYB5b* and whether *VvMYB5b* is involved in the regulation of PA synthesis still needs to be studied.

## 4. Conclusions

The prediction results of miR828a and the target genes found that *VvMYBPA1* is the target gene of vvi-miR828a. During the veraison period (11 wpf), the PA content of ‘Benitaka’ peels was significantly higher than that of ‘Italia’ peels. The transient expression of miR828a during the veraison period of strawberry fruits can significantly slow down their coloration. The results of the transient expression of miR828a and *VvMYBPA1* in grape leaves and qRT-RCR testing showed that miR828a inhibits the expression of PA synthesis-related genes. The above results illustrate that vvi-miR828a can negatively regulate the synthesis of proanthocyanidins in grape berries by inhibiting the expression of the *VvMYBPA1* gene.

## 5. Materials and Methods

### 5.1. Plant Materials

The ‘Italia’ and ‘Benitaka’ grapes were obtained from the national grape germplasm grapeyard (Zhengzhou Fruit Research Institute, Chinese Academy of Agricultural Sciences, Zhengzhou, China). Grape peels were collected before the veraison period (10 weeks after flowering, 10 wpf) and after the veraison period (11 weeks after flowering, 11 wpf), and were named It10, It11, Be10, and Be11, respectively. Two vines and 10 clusters from each cultivar were selected for the current study. A total of 5~10 berries (clean and healthy with uniform size) were collected from the upper, middle, and lower clusters, then quickly and immediately put into liquid nitrogen and transferred to the laboratory to store them in a −80 °C refrigerator for the extraction of total RNA and subsequent experiments. The peels of ‘Italia’ and ‘Benitaka’ berries collected during the 10 wpf and 11 wpf periods (Figure 8) were used as small RNA sequencing (seq) materials to generate 4 independent small RNA-seq libraries.

### 5.2. Total RNA Extraction

Total RNA was extracted from sample tissues using a kit (Solebao Biotechnology Co., Ltd., Shanghai, China); a Nanodrop2000 (Thermo Fisher Scientific, Waltham, MA, USA) was used to detect the concentration and purity of the extracted RNA; agarose gel electrophoresis was used to detect RNA integrity; and Agilent5300 was used to determine the RQN value. Requirements for single library construction: the total amount of RNA was ≥ 1 ug, the concentration of RNA was ≥50 ng/μL, RQN was >7, and OD260/280 was between 1.8 and 2.2.

### 5.3. RNA Library Preparation

Isolate and purify the above-mentioned mRNA, use T4 RNA ligase to add adapter sequence 1 to the 3′ end, and then use T4 RNA ligase to add adapter sequence 2 to the 5′ end. Use the reverse transcription kit (TruSeqTM RNA sample prep Kit) to reverse transcribe the mRNA by using random primers; the RNA which were connected with the adapter are used as a template to reverse synthesize one-strand cDNA and then perform second-strand synthesis to form a stable double-stranded structure to construct a cDNA library. The sequencing primers are used to perform PCR amplification (11–12 cycles) to enrich the library concentration. According to the different characteristics of the length distribution of miRNA, target fragments are recovered by cutting the PAGE gel (6% Novex TBE PAGE gel, 1.0 mm, 10 well). After single-strand circularization of library DNA and the removal of uncirculated sequences and purification, DNA nanoballs (DNB) are prepared and sequenced using the Illumina HiSeq 4000 (300 cycles) system (Majorbio Biomedical Technology Co., Ltd. Company, Shanghai, China).

### 5.4. RNA Sequencing and Comparison

The sequencing image signals were converted into text signals through CASAVA base calling and stored in fasta format as raw data. The data of each sample was separated according to the index sequence for subsequent analysis.

Fastx_toolkit (Version 0.0.14) software (http://hannonlab.cshl.edu/fastx_toolkit/ (accessed on 5 April 2022) was used for quality control. To obtain clean reads, the sequence data were filtered and removed adapter sequences containing poly-N sequence low-quality reads and sequences with reads length < 18 nt. HISAT2 (Version 2.1.0) (https://daehwankimlab.github.io/hisat2/ (accessed on 5 April 2022) software was utilized to compare clean reads with the grape 12× reference genome (https://plants.ensembl.org/Vitis_vinifera/Info/Index?db=core (accessed on 5 April 2022) and to perform sequence comparison. StringTie (Version 1.3.3b) (https://ccb.jhu.edu/software/stringtie/index.shtml?t=example (accessed on 5 April 2022) software was used to assemble transcripts from the clean read data for each sample after quality control. After the completion of reference genome annotation, the read alignment rate was generally higher than 65% (total mapped reads). By searching in the Rfam11.0 database, the filtered sequences were annotated. The Repeat, rRNA, tRNA, snRNA, and snoRNA were first determined, and then the remaining sequences (sequences with a length of 18~25 nt) were used to identify known miRNAs and predict new miRNAs (unknown miRNAs).

### 5.5. Identification of Known miRNAs and Prediction of New miRNAs

The reads of the reference genome with the miRBase and Rfam databases were compared to obtain known miRNA and ncRNA annotation information, respectively. The miRNA identification and prediction process includes three parts: identification of known miRNAs, prediction of new miRNAs, and sRNA statistics. (1) Identification of known miRNAs: compared the mapped reads to the reference genome with the miRNA precursor and mature sequences in the miRBase database, counted the detail of the matched miRNAs in each sample, and predicted its secondary structure. (2) Prediction of new miRNAs: miReap software (Version 3)(http://mireap.sourceforge.net/ (accessed on 11 April 2022) was used to predict new miRNAs from reads that do not have annotation information. sRNA (that was unable to align) was aligned with Rfam and miRBase to the reference the genome and intercepted its near sequences. Rfam and miRBase software was used to predict the secondary structure of pre-miRNAs as well. Based on the prediction results, dicer digestion site information, energy values, and other features were used to filter and identify new miRNA. (3) sRNA statistics: the Rfam database (http://rfam.xfam.org/ (accessed on 20 April 2022) was used to annotate reads of known miRNAs that were not aligned—as well as filtered ribosomal RNA (rRNA), transfer RNA (tRNA), nuclear small RNA (snRNA), small nucleolar RNA (snoRNA), and other ncRNAs and repetitive sequences—counting the types and numbers of these sequences to obtain unannotated reads containing potential miRNAs.

### 5.6. Differential Expression Analysis of miRNAs

After gene expression analysis, the read count of transcripts was carried out; the expression level of each transcript was calculated based on the average number of transcripts per million, corresponding to a specific gene or transcript in the sample (TPM) method. RSEM (http://deweylab.biostat.wisc.edu/rsem/ (accessed on 22 April 2022) was used to quantify gene abundance. DEGseq was utilized to perform differential gene expression analysis, and the screening parameters were |log_2_FC| > 1, Q value (*p*-adjust value) ≤ 0.05. The Majorbio cloud platform (Majorbio Biomedical Technology Co., Ltd., Shanghai, China) was used to get TPM values and DEG miRNAs (differentially expressed miRNAs) data.

### 5.7. Target Gene Prediction and Enrichment Analysis of miRNA

The online software (Version 2, 2017 release)PsRNATarget (https://www.zhaolab.org/psRNATarget/analysis?function=1 (accessed on 18 June 2022) was used to predict target genes of differentially expressed miRNAs; the maximum expected value was set to 5. Enrichment analysis of predicted target genes was performed through Gene Ontology (GO) and Kyoto Encyclopedia of Genes and Genomes (KEGG) analysis. The software Goatools (Version 0.6.5) and the Fisher method were used to test *p* value (*p*-adjust value) < 0.05.

### 5.8. Dual Luciferase Reporter Gene Assay to Verify Target Genes

A dual-luciferase reporter gene transient expression system was used to study the relationship between miRNA and target genes. Using vvi-miR828a as the effector and the corresponding target gene *VvMYBPA1* as the reporter, we explored the regulatory effect of vvi-miR828a on its target gene. Using ‘Italia’ grape cDNA as a template, the target sequence of vvi-miR828a and its flanking sequences (150 bp fragments upstream and downstream of the target sequence) were amplified and cloned into the intermediate vector pEASY^®^-Blunt. At the same time, the target sequence was artificially changed, and three base CTAs were added to the 10~11th bases of the target sequence. The sequence was synthesized by Sangon Biotech; it was then cloned into the intermediate vector pEASY^®^-Blunt. After verification by DNA sequencing, the plasmid was extracted and cloned into the pGreenII 0800-LUC vector (at the 3’ end of the LUC gene) that had been double digested with EcoRI and HindIII to complete the construction of the vector. The *VvMYBPA1* CDS sequence was subsequently cloned using the ‘Italia’ grape cDNA template. The *VvMYBPA1* CDS sequence was constructed into the pBI-121 vector through a homologous recombination method. It was then transformed into *E. coli* competent Trans5α, and positive clones were selected after overnight culture on a resistant medium. After bacterial liquid PCR and sequencing verification, the positive bacterial liquid was saved. Finally, a plasmid extraction kit (TransGen Biotech Co., Ltd., Beijing, China) was used to extract the plasmid from the positive bacterial fluid, and the fusion plasmid was transformed into an Agrobacterium GV3101 (pSoup) strain by the freeze–thaw method for further experimentation.

The cultured Agrobacterium single colony was activated in a LB liquid medium containing 50 ng/µL Kan/Spec and 25 ng/µL Rif. After collecting the bacterial liquid, a centrifugation step was carried out, the supernatant was discarded, and the bacteria dried. The bacteria were resuspended in the preprepared infection solution and incubated for about 4 h under dark conditions. During the induction period, a mixture of effectors and reporters (9:1) was prepared and injected back into the selected tobacco leaves. After the injection, the tobacco seedlings were cultured in dark conditions for 12 h and then transferred to the conventional culture conditions to continue culturing for 2 d. After the culture, the corresponding dual-luciferase activity was measured using a GloMax^®^ Navigator System (Promega, Madison, WI, USA). After taking the corresponding number of tobacco leaf samples, they were ground into a powder form and treated with a lysis stock solution. The solution was centrifuged at 6000 rpm for 5~10 min, and the supernatant was used for dual-luciferase activity measurement. Four biological replicates were performed for each combined sample and protected from light during the determination. The supernatant was transferred to the enzyme plate and immediately the luciferase reaction solution was added to the LUC value. Stop & Glo reaction solution was added to the same mixture to observe the REN value. The LUC/REN ratio was calculated, and SPSS v17.0 software was used for statistical analysis. A T-test was used to determine the significant difference (*p* < 0.05).

### 5.9. Construction of Transient Overexpression Vector

The method of cloning the vvi-miR828a precursor stem-loop sequence using the cDNA of ‘Italia’ grapes as a template was as follows: the PCR reaction system included 1 μL of each forward and reverse primer, 1 μL of cDNA template, 22 μL of ddH_2_O, 25 μL of high-fidelity enzyme, and 50 μL for the total system. The PCR program was set as follows: pre-denature at 98 °C for 1 min, then perform 30 cycles of the following steps: denaturation at 98 °C for 30 s, annealing at 60 °C for 15 s, extension at 72 °C for 30 s, and finally store at 4 °C. The purified target fragment pre-miR828a was then ligated into the intermediate vector pEASY^®^-Blunt (TransGen Biotech Biotechnology Co., Ltd., Beijing, China) vector using T4 DNA ligase. The ligated products were transformed into *E. coli* to confirm positive clones. Positive clones were inoculated into LB liquid medium containing Kan, and after culturing for 12 h at 200 rpm and 37 °C, the pEASY-pre-miR828a plasmid was extracted using the EasyPure plasmid extraction kit (TransGen Biotech Biotechnology Co., Ltd., Beijing, China). Using the extracted plasmid as a template, the seamless cloning kit (Beyotime Biotechnology Co., Ltd., Shanghai, China) was used to clone it into the pBI-121 vector, which was driven by the CaMV35S strong promoter to generate the recombinant overexpression vector pBI-121-OX-miR828a. Finally, OX-miR828a was used as a template to transform it into Agrobacterium competent GV3101 for later use. The cDNA of ‘Italia’ grapes was used to clone the CDS sequence of *VvMYBPA1*, and the subsequent method is the same as 1.8.

### 5.10. Transient Overexpression Test

After selecting a single colony from the constructed pBI-121-OX-miR828a and pBI-121-*VvMYBPA1* vectors, it was inoculated in 20 mL LB liquid culture medium containing 50 ng/mL Kan and 25 ng/mL Rif and incubated at 28 °C at 200 rpm to inactivate the strains overnight. When the OD600 of the bacterial solution reached approximately 0.8~1.0, the bacterial solution was centrifuged at 6000 rpm for 5~10 min to collect the bacterial cells. Subsequently, the collected bacteria were re-suspended in a suspension containing 200 mM acetosyringone, 200 mM magnesium chloride, and 200 mM MES (pH 5.7), and the optical density was adjusted to 1.0 and placed in the dark at room temperature for 4 h. The 2nd to 3rd spine grape leaves were selected below the growth. Three leaves from each group were selected for treatment. The vacuum infiltration method was utilized to infect the grape leaves. The leaves were removed after 10 min of infection and the leaf surface was gently wiped with filter paper to remove excess infection fluid. The petiole was covered with soaked absorbent cotton and the leaf placed in the LED light incubator. Samples were collected on the 4th day post treatment for subsequent analysis. The strawberry cultivar ‘Benihoppe’ (*Fragaria* × *ananassa*) was grown in a glass greenhouse and used in the transient overexpression experiment. A quantity of 1 mL extract from the prepared bacterial liquid was injected in the selected strawberry fruits before the veraison stage. After injection, fruit phenotype changes were continuously observed every day for 7 days.

### 5.11. Real-Time Fluorescence Quantitative PCR (qRT-PCR) Verification

Verification of miRNA828a expression by qRT-PCR: extract from grape peels was used to perform reverse transcription of miR828a using the stem-loop primer method as described by Shi [49]; after the reverse transcription, a qRT-PCR test was performed using SYBR Green Premix Pro Taq HS qPCR Kit on a fluorescence quantitative PCR instrument (Hangzhou Bioer Technology Co., Ltd., Hangzhou, China). qRT-PCR conditions were as follows: pre-denaturation at 95 °C for 1 min; denaturation at 95 °C for 15 s, extension at 60 °C for 1 min, 40 cycles. U6 (primer: F: 3’-CCGATAAAATTGGAACGATACAGAG-5’; R: 3’TCGATTTGTGCGTGT-CATCCT-5’) was used as housekeeping miRNA. All qRT-PCR reactions were performed with 3 technical replicates. SPSS Statistics 20 software was used to analyze the significant differences in expression levels between different samples. qRT-PCR verification of genes related to anthocyanin and proanthocyanidin synthesis in transient overexpression and control RNA samples was performed. The expression levels of genes related to anthocyanin and proanthocyanidin synthesis *VdPAL, VdCHS, VdCHI, VdDFR, VdMYB5b, VdANR*, and *VdMYBPA1* in the RNA of transient overexpression and control samples were analyzed using qRT-PCR, and *VdActin* was used as the housekeeping gene. The PCR reaction procedure refers to the instructions of the SYBR Green Premix Pro Taq HS qPCR Kit: pre-denaturation at 94 °C for 30 s; denaturation at 94 °C for 5 s, annealing at 60 °C for 15 s, extension at 72 °C for 10 s, 40 cycles. The relative expression of each gene was calculated using the 2^−ΔΔCt^ method.

## Figures and Tables

**Figure 1 plants-13-01688-f001:**
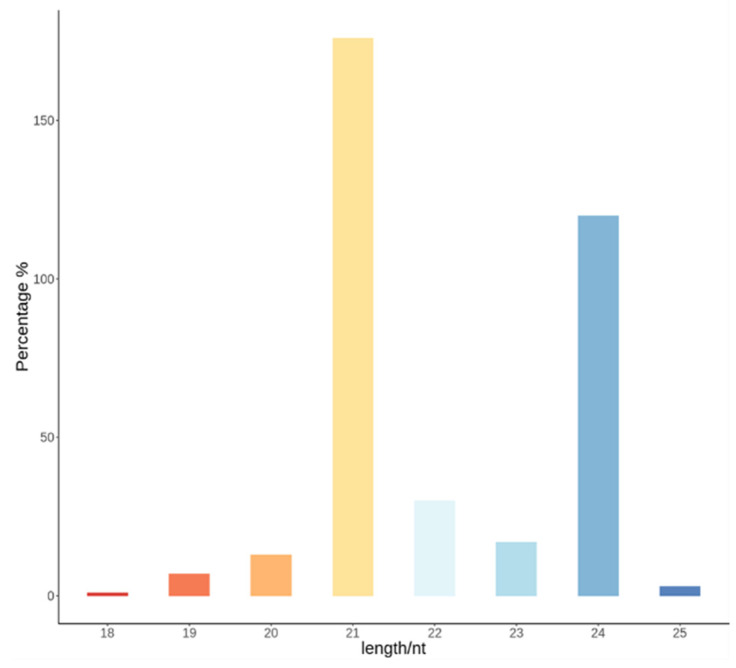
Expression proportions of small RNAs of different lengths in four samples.

**Figure 2 plants-13-01688-f002:**
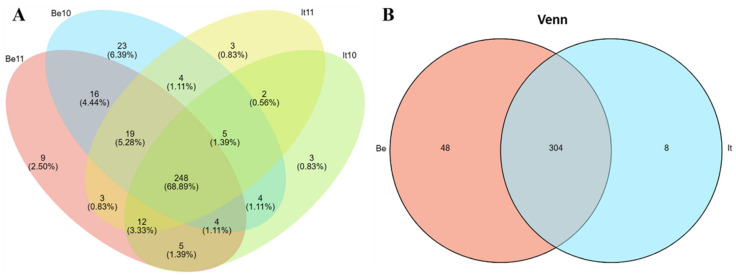
Venn diagram of the number and proportion of miRNA expressions in each sample. (**A**) The expression number and proportion of all identified miRNAs in each sample. (**B**) The number of co-expressed and specifically expressed miRNAs in the ‘Italia’ and ‘Benitaka’ cultivar.

**Figure 3 plants-13-01688-f003:**
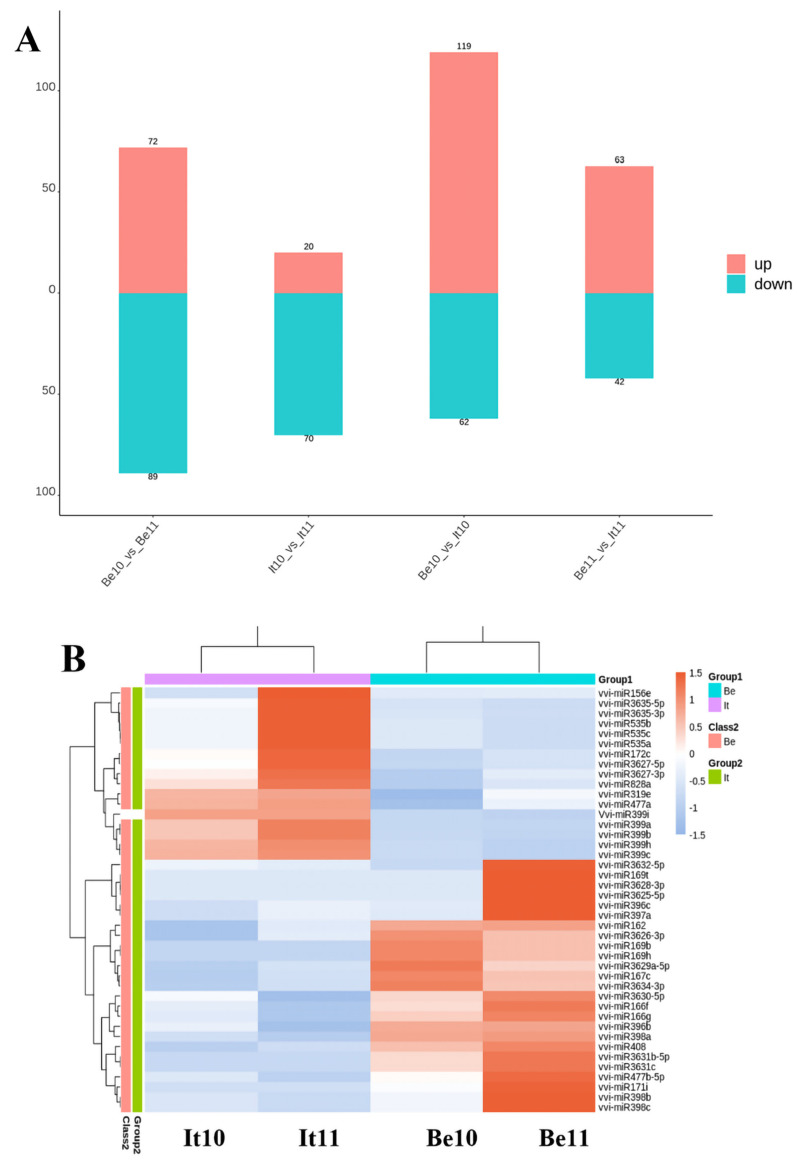
Number and cluster analysis of up-regulated and down-regulated differentially expressed miRNAs. (**A**) Pairwise comparison of differentially expressed miRNAs between different samples. (**B**) Expression clustering heat map of differentially expressed miRNAs in four samples; the red color represents a high miRNA expression, while the blue color represents a low miRNA expression.

**Figure 4 plants-13-01688-f004:**
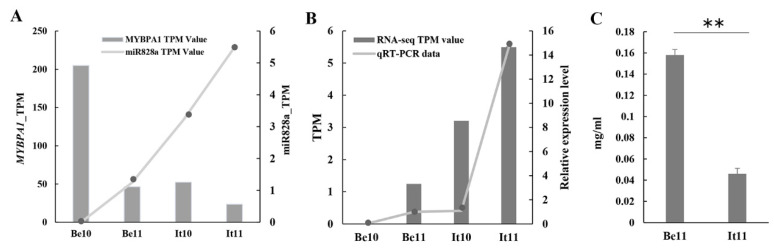
MYBPA1 gene expression, qRT-PCR verification, and proanthocyanidin content. (**A**) VvMYBPA1 and miR828a transcriptome sequencing TPM values. (**B**) qRT-PCR detection of vvi-miR828a expression to verify the accuracy of miRNA sequencing data. (**C**) Proanthocyanidin (PA) content in the peel of ‘Italia’ (It11) and ‘Benitaka’ (Be11) cultivars at 11 weeks post flowering; ** indicates *p* < 0.01.

**Figure 5 plants-13-01688-f005:**
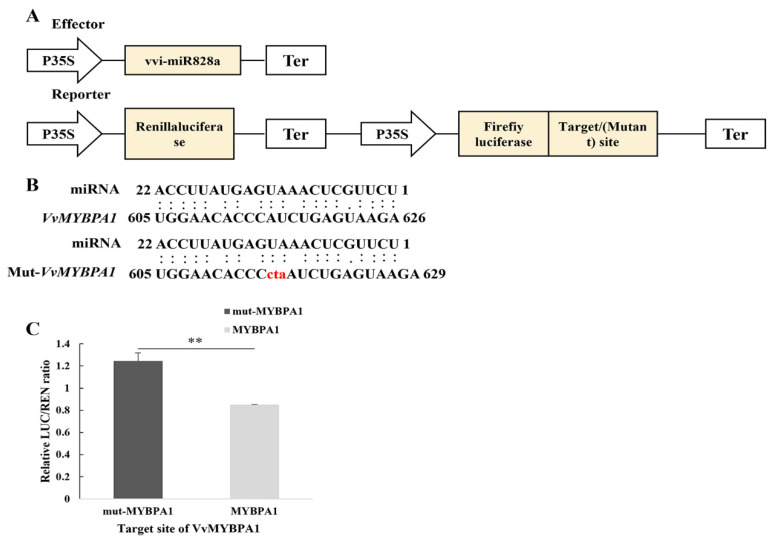
Verification of vvi-miR828a and target gene *VvMYBPA1* using a dual-luciferase assay. (**A**) Schematic diagram of dual-luciferase vector. (**B**) Recognition sites of vvi-miR828a, target gene *VvMYBPA1*, and its mutant sequence mut-*VvMYBPA1*; the red “cta” base indicates the mutant bases. (**C**) Luc/Ren ratio (*MYBPA1* indicates the Luc/Ren ratio of the vvi-miR828a target sequence; mut-*MYBPA1* indicates the Luc/Ren ratio of the vvi-miR828a target sequence mutation; ** indicates *p* < 0.01).

**Figure 6 plants-13-01688-f006:**
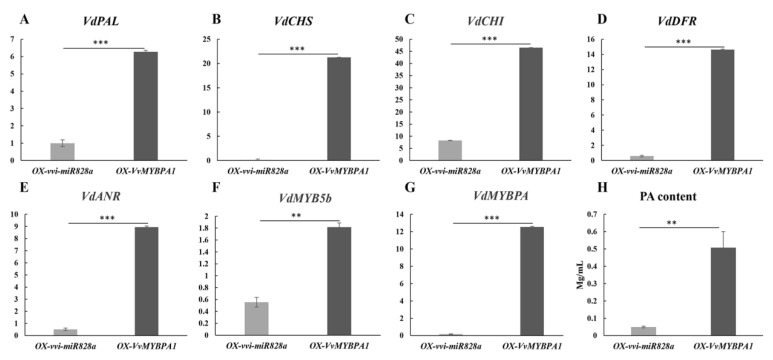
Expression levels of genes related to anthocyanin and proanthocyanidin synthesis after the transient expression of miR828a and *VvMYBPA1*. (**A**–**G**) These represent the expression levels of the *VdPAL*, *VdCHS*, *VdCHI*, *VdDFR*, *VdMYB5b*, *VdANR*, and *VdMYBPA1* genes, respectively, after transiently expressing pre-miR828a and *VvMYBPA1* in grape leaves for 3 days. The gray bar graph (OX-vvi-miR828a) represents the expression levels of anthocyanin and proanthocyanidin synthesis-related genes after the injection of bacterial fluid containing the vvi-miR828a vector; the black bar graph part (*OX-VvMYBPA1*) represents the expression level of related genes after the injection of bacterial fluid containing the *VvMYBPA1* vector. (**H**) PA content after the transient expression of pre-miR828a and *VvMYBPA1* in *Vitis davidii* leaves; ** indicates *p* < 0.01, indicating significant difference, *** indicates *p* < 0.001, indicating highly significant difference.

**Figure 7 plants-13-01688-f007:**
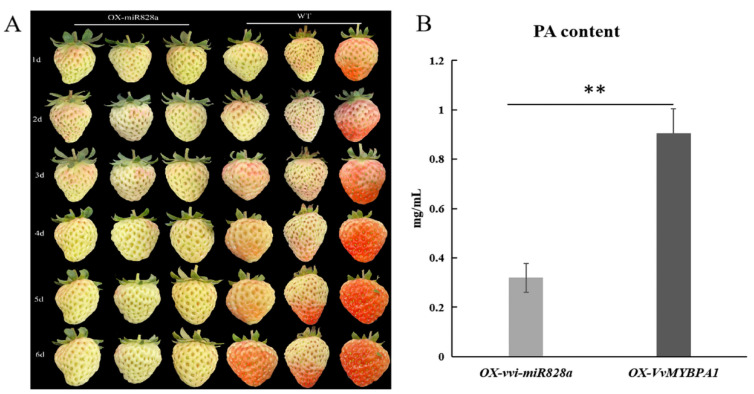
Transient overexpression of vvi-miR828a in strawberry fruits at the veraison stage. (**A**) Phenotypes were observed 1~6 days after the transient overexpression of vvi-miR828a. The three columns on the left represent strawberry fruits 1~6 days after the injection of a bacterial solution that contains the vvi-miR828a vector. The three columns on the right represent strawberry fruits 1~6 days after the injection of a bacterial solution that contains an empty vector. (**B**) Proanthocyanidin content in strawberry fruits after the transient overexpression of vvi-miR828a and *VvMYBPA1*; ** indicates *p* < 0.01, indicating significant difference.

**Figure 8 plants-13-01688-f008:**
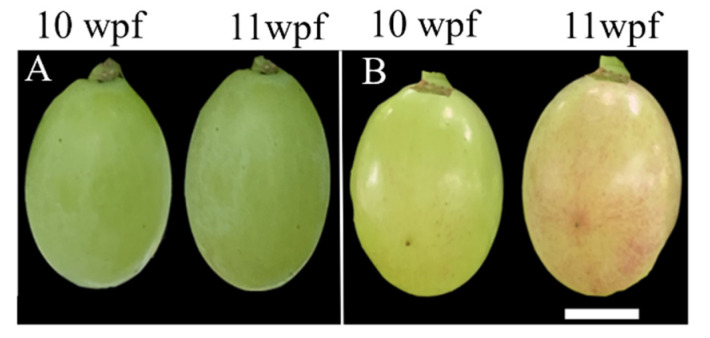
Fruits of ‘Italia’ and ‘Benitaka’ before and after the veraison period. (**A**) representative single fruits of ‘Italia’ at 10 weeks (10 wpf) and 11 weeks (11 wpf) post flowering. (**B**) representative single fruits of ‘Benitaka’ at 10 weeks and 11 weeks post flowering; scale bar length is equal to 6 mm.

**Table 1 plants-13-01688-t001:** Small RNA sequencing data of four samples.

Type	It10	It11	Be10	Be11
Known miRNA	1,461,607 (21.93%)	1,349,557 (22.46%)	1,686,465 (21.27%)	1,441,903 (20.83%)
Novel miRNA	326,182 (4.89%)	315,922 (5.26%)	362,619 (4.57%)	365,986 (5.29%)
rRNA	1,880,625 (28.21%)	1,140,941 (18.99%)	1,635,203 (20.62%)	1,242,239 (17.94%)
tRNA	46,149 (0.69%)	17,279 (0.29%)	28,357 (0.36%)	25,122 (0.36%)
snoRNA	8354 (0.13%)	6624 (0.11%)	18,816 (0.24%)	9239 (0.13%)
snRNA	527 (0.01%)	660 (0.01%)	1341 (0.02%)	1151 (0.02%)
Repbase	3117 (0.05%)	3035 (0.05%)	3919 (0.05%)	2856 (0.04%)
Exon	680,058 (10.2%)	535,576 (8.92%)	722,475 (9.11%)	720,531 (10.41%)
Intron	403,736 (6.06%)	423,030 (7.04%)	533,595 (6.73%)	557,451 (8.05%)
Unknown	1,856,008 (27.84%)	2,214,860 (36.87%)	2,937,031 (37.04%)	2,556,046 (36.92%)
Total	6,666,363	6,007,484	7,929,821	6,922,524

Note: Novel miRNA, newly predicted miRNA; rRNA, ribosomal RNA; tRNA, transfer RNA; snoRNA, small nucleolar RNA; snRNA, small nuclear RNA; unknown, unannotated fragment.

**Table 2 plants-13-01688-t002:** TPM values and functional annotations of target genes predicted by vvi-miR828a.

miRNA and Target Gene Name	Be10 (TPM)	It10 (TPM)	Be11 (TPM)	It11 (TPM)	Annotation from NCBI
vvi-miR828a	0	3.2054	1.2463	5.4922	
VIT_08s0007g04830	0	0.31	0	0	MYB-related protein
VIT_15s0046g00170	204.97	52.37	46.43	23.55	MYBPA1 protein
VIT_04s0008g01470	0.66	3.55	24.74	8.61	Probable WRKY transcription factor 50
VIT_09s0070g00240	87.97	19.2	22.24	10.81	Cinnamoyl-CoA reductase 1
VIT_09s0002g00800	0	0.21	0	0.11	Zinc finger CCCH domain-containing protein
VIT_15s0048g02120	0.61	3.63	0.77	1.61	Transcriptional activator MYB
VIT_04s0008g00660	9.17	32.02	17.65	41.37	Protein early flowering 3
VIT_14s0006g01280	0.76	0	0	0	MYBA7-R2R3 MYB transcription factor
VIT_06s0004g00570	9.66	12.69	18.48	13.44	*MYB5b*
VIT_08s0007g07230	27.14	19.46	28.73	24.12	MYBCS1-MYB transcription factor
VIT_17s0000g06190	175.15	105.52	133.45	87.57	MYB30-R2R3 MYB30 transcription factor

Note: Red font represents the expression of vvi-miR828a and the most obviously regulated target gene/protein MYBPA1.

## Data Availability

The raw sequencing data has been uploaded to NCBI; The login number is: PRJNA1121963.

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
