# Peer review of "Regulatory Mechanism of Proanthocyanidins in Grape Peels Using vvi-miR828a and Its Target Gene VvMYBPA1"

_plants, 2024, doi:10.3390/plants13121688_

Round 1
Reviewer 1 Report
Comments and Suggestions for Authors
See attachment.

The English expression in the entire manuscript needs to be extensively revised. There are so many run-on sentences and other grammar issues throughout the manuscript.
Author Response
Dear reviewers:
Thank you very much for your suggestions for our modifications.
Reply to the first question: The issue you mentioned does exist in our two results of 2.7 and 2.8 part. In section 2.7, we performed qRT-PCR for each gene in Figure 6A-G on the wild type grape leaves in the early stage as well, the results showed that the gene expression level of the wild-type control was lower than that after injection agrobacterium bacteria solution of OX-vvi-miR828a and OX-VvMYBPA1, it is difficult for us to explain why this causing these results currently (Therefore, this result is not presented in Section 2.7). In Section 2.8, we did not conduct the strawberry transient transformation to test the content of PA in our control (CK). Unfortunately, there are currently no strawberries for supplementary testing in this section at this summer time in our province. Therefore, we chose OX-vvi-miR828a and OX-VvMYBPA1 for comparison, combined with the results of 2.6, 2.7, and 2.8, to a certain extent, it can be explained that the expression level of injection of OX-vvi-miR828a is lower than that of injection of OX-VvMYBPA1 agrobacterium bacteria solution, it can reflect the regulatory effect of miRNA on proanthocyanidins to a certain extent as well.
Reply to the second question: there are two reasons why we use strawberries for transient expression experiments of miRNA as follows: First, because grapes were in the dormant period during this experiment, there was no suitable test material (grape fruits) to conduct overexpression experiments, therefore, the biological function of miRNA could not be verified in grapes at that time. Secondly, we read some paper and conducted bioinformatics analysis of the studied miRNA. miR828 is a highly conserved gene in plants, so we chose strawberry, a "color transfer model species" in this study for transient transformation test verification.
We checked 2 paper that used strawberries for transient functional verification in grape research as follows:
(1) The Effect of Ethylene on the Color Change and Resistance to Botrytis cinerea Infection in ‘Kyoho’ Grape Fruits.
(2) Chitinase family genes in grape diferentially expressed in a manner specifc to fruit species in response to Botrytis cinerea.
Our research group is also trying using grape as a transgenic system, and our subsequent research will focus on grape genetic functional verification.
Thank you again for your valuable review suggestions.
Best regards,
Yue Lingqi & Yanshuai Xu
Reviewer 2 Report
Comments and Suggestions for Authors
Line 98: change ; for a dot “.”
Results:
Line 205: =
Lines 220-231: Include a supplementary table for this information
Line 238: Mentioned the parameter you used for mentioned that are differentially express, logFc and FDR?
Line 269: obvious strong negative correlation?
Line 303: reduced . On the contrary
Methods:
Line 446: two vines, not two trees
Line 448: quickly immediately put them into liquid nitrogen
Line 470: was
Lines 483 – 495:
The current writing lacks some crucial information and has an illogical order in the software usage. Here's a revised version with suggestions for improvement:
Software Order:
Currently, StringTie is used before Hisat2, which is incorrect. StringTie is for assembling transcripts from aligned reads, while Hisat2 aligns reads to the reference genome. The correct order should be Hisat2 first, followed by StringTie.
Missing information:
Software versions: Specify the versions of the software used (Trimmomatic, Hisat2 and StringTie) for reproducibility.
Filter parameters: Mention the specific parameters used in Trimmomatic to filter the reads (e.g. minimum quality, minimum length, etc.).
Conclusion:
The study provides compelling evidence for vvi-miR828a negatively regulating proanthocyanidin synthesis in grapes by inhibiting VvMYBPA1. The correlation between lower miR828a activity and higher proanthocyanidin content in 'Benitaka' grapes during ripening further supports this conclusion. However, caution is advised when extrapolating to strawberries. While vvi-miR828a overexpression delays strawberry coloration, further research is needed to confirm a direct link with proanthocyanidin content and explore if this mechanism is conserved across different plant families
There are some instances of misplaced full stops and commas. A careful review of punctuation will ensure that the text flows naturally and is easy to understand.
Author Response
Dear reviewers:
Thank you very much for your suggestions for our modifications.
Your valuable suggestions, we have made revisions. In addition, we will communicate with the editor to supplement the documents.
Reply to the first question: The issue you mentioned does exist in our two results of 2.7 and 2.8 part. In section 2.7, we performed qRT-PCR for each gene in Figure 6A-G on the wild type grape leaves in the early stage as well, the results showed that the gene expression level of the wild-type control was lower than that after injection agrobacterium bacteria solution of OX-vvi-miR828a and OX-VvMYBPA1, it is difficult for us to explain why this causing these results currently (Therefore, this result is not presented in Section 2.7). In Section 2.8, we did not conduct the strawberry transient transformation to test the content of PA in our control (CK). Unfortunately, there are currently no strawberries for supplementary testing in this section at this summer time in our province. Therefore, we chose OX-vvi-miR828a and OX-VvMYBPA1 for comparison, combined with the results of 2.6, 2.7, and 2.8, to a certain extent, it can be explained that the expression level of injection of OX-vvi-miR828a is lower than that of injection of OX-VvMYBPA1 agrobacterium bacteria solution, it can reflect the regulatory effect of miRNA on proanthocyanidins to a certain extent as well.
Reply to the second question: there are two reasons why we use strawberries for transient expression experiments of miRNA as follows: First, because grapes were in the dormant period during this experiment, there was no suitable test material (grape fruits) to conduct overexpression experiments, therefore, the biological function of miRNA could not be verified in grapes at that time. Secondly, we read some paper and conducted bioinformatics analysis of the studied miRNA. miR828 is a highly conserved gene in plants, so we chose strawberry, a "color transfer model species" in this study for transient transformation test verification.
We checked 2 paper that used strawberries for transient functional verification in grape research as follows:
(1) The Effect of Ethylene on the Color Change and Resistance to Botrytis cinerea Infection in ‘Kyoho’ Grape Fruits.
(2) Chitinase family genes in grape diferentially expressed in a manner specifc to fruit species in response to Botrytis cinerea.
Our research group is also trying using grape as a transgenic system, and our subsequent research will focus on grape genetic functional verification.
Thank you again for your valuable review suggestions.
Best regards,
Yue Lingqi & Yanshuai Xu

Round 2
Reviewer 1 Report
Comments and Suggestions for Authors
The authors addressed my concerns regarding the scientific questions present in the first version.
Comments on the Quality of English LanguageThe English language issues are almost not revised at all.
Author Response
Dear reviewers,
Hello!
Thank you very much for your valuable advice. We talked to René about language changes, and she suggested that we change the language after we have finished changing the questions.
Thank you for your understanding. Have a great day!

Reviewer 2 Report
Comments and Suggestions for Authors
Please check the following lines:
Lines 220-231: Include a supplementary table for this information Author reply: Done
Where is included?
Lines 491: Stringtie function is not very clear.... maybe:
StringTie (Version 1.3.3b) software was used to assemble transcripts from the clean reads data for each sample after quality control
Please don't forget to indicate in the text where you upload your miRNA and transcriptomic RNA-seq reads.
Author Response
Dear reviewers,
Hello!
Thank you very much for your valuable questions!
Lines 220-231: We have now supplemented Schedule 1.
Lines 491:The question of the place has been revised.
We immediately upload the miRNA-seq to the NCBI database. The SRA number will be indicated in the article later.
Thank you for your understanding. Have a great day!
